REGISTERED REPORT PROTOCOL

# Balance rehabilitation with a virtual reality protocol for patients with hereditary spastic paraplegia: Protocol for a clinical trial

**Bianca Simone Zeigelboim**[1¤a], **Maria Renata José**[1¤a], **Geslaine Janaina Bueno dos Santos**[2☯¤b], **Maria Izabel Rodrigues Severiano**[2¤c], **Hélio Afonso Ghizoni Teive**[3☯], **José Stechman-Neto**[1¤a], **Rosane Sampaio Santos**[1¤a], **Cristiano Miranda de Araújo**[1¤a], **Bianca Lopes Cavalcante-Leão**[1¤a]*

**1** UTP- Universidade Tuiuti do Paraná, Curitiba, Paraná, Brazil, **2** IFPR - Instituto Federal do Paraná, Curitiba, Paraná, Brazil, **3** UFPR – Universidade Federal do Paraná, Curitiba, Paraná, Brazil

☯ These authors contributed equally to this work.
¤a Current address: R: Sydnei Antonio Rangel Santos, Curitiba - Paraná, Brasil
¤b Current address: R: General Carneiro, Curitiba - Paraná, Brasil
¤c Current address: R. João Negrão, Curitiba - Paraná, Brasil
* blcleao@gmail.com

## Abstract

### Background

Neurodegenerative diseases are sporadic hereditary conditions characterized by progressive dysfunction of the nervous system. Among the symptoms, vestibulopathy is one of the causes of discomfort and a decrease in quality of life. Hereditary spastic paraplegia is a heterogeneous group of hereditary degenerative diseases involving the disorder of a single gene and is characterized by the progressive retrograde degeneration of fibers in the spinal cord.

### Objective

To determine the benefits of vestibular rehabilitation involving virtual reality by comparing pre intervention and post intervention assessments in individuals with hereditary spastic paraplegia.

### Methods

In this randomized controlled clinical trial from the Rebec platform RBR-3jmx67 in which allocation concealment was performed and the evaluators be blinded will be included. The participants will include 40 patients diagnosed with hereditary spastic paraplegia. The interventions will include vestibular rehabilitation with virtual reality using the Wii® console, Wii-Remote and Wii Balance Board (Nintendo), and the studies will include pre- and post intervention assessments. Group I will include twenty volunteers who performed balance games. Group II will include twenty volunteers who performed balance games and muscle strength games. The games lasted from 30 minutes to an hour, and the sessions were performed twice a week for 10 weeks (total: 20 sessions).

**Data Availability Statement:** All relevant data from this study will be made available upon study completion.

**Funding:** The author(s) received no specific funding for this work.

**Competing interests:** The authors have declared that no competing interests exist.

## Results

This study provides a definitive assessment of the effectiveness of a virtual reality vestibular rehabilitation program in halting the progression of hereditary spastic paraplegia, and this treatment can be personalized and affordable.

## Conclusion

The study will determine whether a vestibular rehabilitation program with the Nintendo Wii® involving virtual reality can reduce the progressive effect of hereditary spastic paraplegia and serve as an alternative treatment option that is accessible and inexpensive.

Rebec platform trial: RBR-3JMX67.

## Introduction

Neurodegenerative diseases are sporadic hereditary conditions characterized by progressive dysfunction of the nervous system. Such conditions are generally associated with atrophy of the affected structures of the central or peripheral nervous system. Among the symptoms, vestibulopathy is one of the causes of discomfort and a decrease in quality of life. Otoneurologic assessments include pathophysiological evaluations of the vestibular system and its relation to the central nervous system, with an emphasis on the vestibulo-oculomotor, vestibulo-cerebellar, vestibulospinal and cervico-vestibular proprioception interrelationships [1].

Balance is essential to maintaining an erect posture and requires information processed by the visual, vestibular and proprioceptive systems. The proper diagnosis of a balance disorder and adequate rehabilitation are essential for avoiding instability, the loss of balance, a floating sensation, vertigo and falls. Vestibular rehabilitation (VR) involving virtual reality training can be a powerful tool for improving balance [2]. VR has been reported to exert a physiologic effect on the vestibular system, and this therapeutic modality based on central mechanisms of neuroplasticity can promote visual stabilization and improve vestibular-visual interactions during head movements as well as enhance static and dynamic balance under conditions that produce conflicting sensory information and diminish individual sensitivity to head movements [3, 4].

Hereditary spastic paraplegia (HSP) is a heterogeneous group of hereditary degenerative diseases involving the disorder of a single gene and is characterized by the progressive retrograde degeneration of long axonal fibers of the cortico spinal tracts of the spinal cord [5–8]. The main clinical manifestation of HSP is a pyramidal pattern of slowly progressive weakness [5]. VR has been proven to be effective in treating individuals with chronic symptoms and involves eye, head and body movement exercises to promote the neuronal plasticity of the central nervous system, thereby promoting the adaptation of deficient or abnormal vestibular impulses. The goal of this multidisciplinary therapy for these patients is to improve their global balance and quality of life as well as restore their level of spatial orientation to be closer to the normal physiological level [9].

Virtual reality can be used therapeutically to treat these patients by promoting stability and improving vestibular-visual interactions as well as enhancing static and dynamic postural stability; by improving balance, virtual reality may help reestablish self-confidence, provide greater independence in the development of daily activities, reduce anxiety and improve social interactions [10]. According to the literature, the benefits of this therapeutic modality include the correction of balance and posture, improved locomotion patterns, improved upper and

lower limb functioning and greater motivation on the part of the patient to perform exercises [11].

Severiano *et al.* [12] studied 16 patients with Parkinson's disease and observed that the virtual games Tightrope Walk and Ski Slalom proved to be the most effective in the studied population. The authors found improvement in symptoms, mainly in balance and gait, increasing patients self-confidence.

Santos *et al.* [13] observed that the Soccer Heading, Table Tilt, Tightrope and Ski Slalom games showed significant results in patients with spinocerebellar ataxia. The study was carried out on 28 patients using the Nintendo Wii® equipment, twice a week for 50 minutes, totaling 20 sessions. The authors Lee *et al.* [14] used video game exercises with Sony PlayStation® VR equipment on a 27-year-old stroke survivor. The sessions lasted 30 minutes, 3 times a week, totaling 18 sessions, and the authors observed that the patient obtained 14 points more than the pre-assessment score with 34 points on the Motor Evaluation Scale; 16 points more than the pre-assessment score with 48 points on the Berg Balance Scale; 6.85sec less than the pre-assessment score with a score of 13.58 seconds on the Timed Up and Go test; 5 points more than the pre-assessment score with 13 points on the Tinetti Balance Scale; 5.36 seconds less than the pre-assessment score, with a score of 8.15 sec on the 10 Meter Walk Test; 4 points more than the pre-assessment score with 10 points on the Tinetti Gait Scale and 10 points more than the pre-assessment score with 21 points on the Dynamic Gait Index. This case report suggests that training using a full immersion virtual reality video game may be an effective method to improve motor function, balance and gait in a young stroke survivor. The authors Smits *et al.* [15] refer to the importance of virtual rehabilitation in post-COVID-19 patients, particularly those who have physical rehabilitation needs. The potential of virtual reality linked to rapid and personalized rehabilitation can be a solution to the imminent increase in demand for rehabilitation after COVID-19. Virtual rehabilitation can take the user through computer generated visuals to a realistic and immersive multisensory environment. Immersion of virtual reality can increase adherence to therapy and can distract the patient from fatigue and anxiety. We argue that the incorporation of rehabilitation on virtual platforms would help to stimulate the spread of therapy for both post-COVID-19 patients and other patients of different etiologies with similar rehabilitation needs in the future.

Bruin *et al.* [16] described the advantages of physical exercises based on virtual reality games over conventional balance training; for example, the scenarios and therapeutic protocols can be adapted according to needs and interests of the patient, enabling gains in balance and motor coordination as well as promoting motor learning through changes in the cerebral architecture, which contributes to increasing patients in dependence and motivation to exercise. No randomized or nonrandomized studies related to VR with virtual reality in patients with HSP have been published in the PubMed electronic databases, including Medline, Scielo, Embase, Scopus and Web of Science.

Therefore, the aim of the proposed study was to verify the benefits of VR involving virtual reality by comparing the results of pre intervention and post intervention evaluations in individuals with HSP.

## Materials and methods

### Funding

The primary sponsors of this study, initiated by the investigators, are the authors of the study, and no outside funding was received.

## Design

A randomized, controlled, clinical trial will be conducted in accordance with the ethical principles governing research involving human subjects stipulated in Resolution 466/2012 of the Brazilian National Board of Health. The study received approval from the Human Research Ethics Committee of the *Faculdade Evangélica Mackenzie do Paraná* (process number: 37083714.0.0000.0103; certificate number: 3.580.973), and the protocol was registered and approved on the Rebec platform, trial RBR-3JMX67. All patients agreed to participate by signing a statement of informed consent.

A member of the team, not directly involved in the research, will be responsible for obtaining the signed consent forms from the patients who were initially considered eligible for the trial.

## Study population

Forty patients diagnosed with HSP will be included. The individual and family histories, neurological clinical examination findings, including encephalography, magnetic resonance imaging, urinary function test and genetic test results, will be used to diagnose HSP. All the patients will be recruited from the Movement Disorders Unit, Neurology Department, Clinical Hospital, Federal University of Paraná, Brazil.

To participate, the volunteers can be either male or female and must meet the following eligibility criteria:

### Inclusion.

- Age equal to or greater than 18 years (without restrictions imposed on the maximum age);

- Patients with a diagnosis of HSP confirmed through clinical and /or laboratory tests;

- Individuals who have the ability to understand the explanations of the present study.

### Exclusion criteria.

- An otologic condition that can affect the vestibular examination findings;

- Dependency on a gait-assistance device;

- The inability to understand simple verbal commands;

- A significant musculoskeletal condition that can affect the assessment and VR outcomes;

- Severe visual impairment or another abnormality that can impede the proposed procedures.

## Randomization

Prior to the intervention, the volunteers will be randomly allocated to two different groups. Randomization will be performed by an independent researcher using a simple lottery system with sealed opaque envelopes immediately after the baseline assessment. The volunteers will be considered participants in the study the moment the envelope is opened. The participants will be randomized to receive the following distinct interventions:

**Group I.** Twenty volunteers will undergo VR with virtual reality (balance games) using the Wii® console, Wii-Remote and Wii Balance Board (Nintendo).

**Group II.** Twenty volunteers will undergo VR with virtual reality (balance games and muscle strength games) using the Wii® console, Wii-Remote and Wii Balance Board (Nintendo).

The games will last from 30 minutes to one hour, and the participants will perform sessions twice a week for 10 weeks (total: 20 sessions).

## Treatment

All participants will receive instructions regarding the rehabilitation procedure. The VR program will involve virtual reality with the use of the Wii Fit Plus®, Wii-Remote and Wii Balance Board (Nintendo). The Wii Balance Board (WBB) is a platform with sensors that detect the position and orientation of the gamer. In some games, the player must perform the same movement he/she would in a real game. VR with virtual reality games will be performed in both groups for 30 minutes twice a week for a total of 20 sessions. Group I will play five balance games. Group II will play the same five balance games and four additional strength games. The games will be selected to favor changes in balance and postural instability. After all the data is collected, to ensure the same rehabilitation exercises are provided for both groups, training with strength exercises will be made available to group I.

The following are the balance games that will be performed on the WBB:

Soccer Heading®

Table Tilt®

Tightrope Walk®

Penguin Slide®

Perfect 10®

The following muscle strengthening exercises were selected to improve balance, which depends on the interactions between vision, vestibular and peripheral signals, central commands and neuromuscular responses:

Single leg extension®

Torso Twist®

Sideways Leg Lift®

Single Leg Twist®

The training phases will be performed simultaneously, avoiding complications and changes that may compromise the validity of training. The assessment will be performed again after the 20 intervention sessions.

Eligible patients will undergo the following assessments and treatment:

**Patient history.**   Patient histories will be taken with an emphasis on otoneurological signs and symptoms.

**Otolaryngologic evaluation.**   All patients will undergo inspection of the external ear canal with the MD Mark II model otoscope to determine whether there are any outer ear obstructions according to the criteria of Mangabeira-Albernaz et al. [17].

**Vestibular assessment.**   Subsequently, the participants will undergo the physiological profile assessment (PPA) and Lafayette dynamometer test and respond to the 10 questionnaires/ scales described below before the rehabilitation sessions (1st assessment) and after the 20th rehabilitation session (2nd assessment) for pre—and post intervention comparisons.

**Physiological profile assessment.** Physiological profile assessment (PPA) is a validated assessment tool for the risk of falls and was developed by Lord, Menz and Tiedemann *et al.* [18]. The PPA directly assesses individuals' sensory motor skills. The score indicates the individual's degree of risk of falls relative to the normative level among individuals in the same age group: $<0$ = low risk, 0–1 = mild risk; 1–2 = moderate risk; and $>2$ = high risk of falls.

**Lafayette dynamometer.** The Lafayette dynamometer [19, 20] is a portable device used for the objective quantification of muscle strength based on the assessment of maximum voluntary isometric contraction [21]. This device can be used on all segments of the body and measures the peak force, duration of the peak force, force (kgf) between selected intervals, mean force, total test time and peak torque.

**Quality of life.** The Quality of Life Group of the World Health Organization (WHO) developed the short version of the WHOQOL-100 called the WHOQOL-Bref [22]. This version is a useful alternative to the long version, as the proposed study will involve several assessment tools. The questionnaire is used to determine the perceptions of individuals about their health and can be used to assess the quality of life of different populations and individuals indifferent situations [23].

**Berg Balance Scale (BBS).** The Brazilian version of the BBS, which was cross-culturally adapted to the Brazilian population by Myamoto *et al.* [24], will be used to determine the risk factors for the loss of independence and falls. The maximum score is 56 points, with higher scores denoting better balance. The score will be analyzed to determine the degree of individuals risk of falls: low, medium, high or 100% risk.

**Activities-Specific Balance Confidence (ABC) scale.** The ABC scale, translated and adapted for the Brazilian population [25], has shown to be of good quality, have discriminant validity, and have good consistency and reliability [26]. It has 16 items for assessing balance during a set of activities of daily living (ADLs) of medium difficulty [27]. The confidence of the participant in performing each ADL is measured by the participant as a percentage point on a scale ranging from 0% (without confidence) to 100% (total confidence). A total greater than 80% corresponds to a high functional level, 50 to 80% corresponds to moderate physical functioning and a score below 50% corresponds to a low functional level [28].

**Vestibular Disorders Activities of Daily Living (VADL) scale.** The VADL scale was developed by Cohen and Kimball [29] to assess the impact of dizziness and imbalance on the performance of activities of daily living among individuals with vestibular disorders. The scale involves 28 activities divided into three dimensions: functional, locomotion and instrumental. The total scores and sub scores of the VADL are determined by the median of the activity scores, with higher scores denoting greater dependence and disability [30]. The VADL was translated from English to Brazilian Portuguese in accordance with the Process of Cross-Cultural Adaptation guidelines [31]. The Brazilian version had adequate reliability and is considered a new assessment tool in the country for investigating functional capacity in individuals with vestibulopathies as well as guiding therapies, particularly vestibular rehabilitation [32].

**Mini Mental State Examination (MMSE).** The MMSE is a fast, short, 30-item measure used to screen for cognitive impairment [33] that addresses spatiotemporal orientation, immediate recall and word evocation, calculation, naming, repetition, executing a command, reading, writing and visuomotor skills [34–36]. The MMSE items were categorized by Folstein *et al.* [33] into five dimensions based on theoretical analyses and experiences in clinical practice. The Brazilian version of the MMSE was developed by Brucki *et al.* [35] for use in Brazil, and the authors proposed rules to standardize its application for the assessment of cognitive loss at follow-ups of individuals with diseases and during the monitoring of their responses to treatments. The assessment tool is referred to as "mini" because it focuses on only the cognitive aspects of mental function, excluding mood and abnormal mental functions [33]. It provides

information on different cognitive variables [37] and contains items grouped into seven categories, each of which is designed to evaluate a specific cognitive function. The total score ranges from 0 points (severe cognitive impairment) to 30 points (best cognitive performance).

**Timed Up-and-Go (TUG) test.** According to Paula, Alves Jr. and Prata [38], the TUG test consists of standing up from a chair without using one's arms, walking three meters, turning around, walking back to the chair and sitting down again. The test begins and ends with the volunteer seated with his or her back against the backrest of the chair. The volunteer begins with the command "go", and the time needed to complete the task is timed with a stopwatch. According to the literature, there is no consensus on the results of this test. However, Guimarães *et al.* [39] established the following guidelines: a TUG test completion time of less than 10 seconds indicates a low risk of falls; that of 10 to 20 seconds indicates a medium risk; and that of more than 20 seconds indicates a high risk of falls.

**Falls Efficacy Scale—International (FES-I).** The FES-I is a self-administered questionnaire designed to evaluate the fear of falling during a set of activities. It is composed of 16 items scored from 1 to 4 points, with higher scores denoting a great fear of falling during a given activity. The FES-I has an internal consistency of 0.96. Moreover, most items enable the differentiation of individuals who have suffered a fall, those who have suffered two or more falls and those who have not fallen [40]. The version validated in the Portuguese language will be used [41].

**Visual Analog Scale (VAS).** The VAS is a one-dimensional scale used to measure pain intensity and consists of a 10-cm horizontal line with zero at one end (accompanied by the expression "no pain") and 10 at the other end ("unbearable pain"). Pain will be classified as mild (1 to 3 cm), moderate (4 to 6 cm) or severe (7 to 9 cm). The absence of pain (score: 0) indicates that the participant has no difficulties performing activities of daily living; mild pain indicates that the participant has the ability to perform activities despite experiencing pain; moderate pain indicates that the participant's ability to perform activities is partially or completely impaired; and severe pain indicates that the participant cannot complete the activities [42].

## Management

The researchers will ensure that the anonymity of the participants is protected, and the data concerning people will remain confidential so that their identities and any kind of identifying information is protected. The clinical records, research instruments or any documents that are used that contain data from the participants will not be identified by the participant's name but rather by a code, even when the information is submitted to regulatory institutions or sponsors. The researchers will protected records of the participants included, and the information on the codes, names and addresses of the participants will remain confidential and available only to the researchers. Two copies of the consent forms will be provided, both of which will be signed by the participants, researchers and participants' guardians (or legal representatives). All files will be stored by the researchers in a secure location in a single folder dedicated to this study. If a participant does not follow these rules, he or she will receive an original copy of the consent form.

There will be no personal expenses for the participant during any stage of the study; instead, all assessments will be free, including the otorhinolaryngological evaluation, PPA and dynamometer test. There will also be no financial compensation for participation. If there are any additional expenses, they will be covered by the research budget. Participation in this study is completely voluntary, and nonparticipation does not imply any changes in the patient's medical follow-up and will not even change the relationship between the team and the patient.

After the patients sign the consent form, they will be able to terminate their participation in the study at any time, if desired, without any consequences related to their treatment or follow-up at the institution.

The results of this research may be presented in at meetings or publications; however, the participants' identity will not be revealed in these presentations.

The risks during the research are minimal or nonexistent, but the assessments and rehabilitation procedures may cause discomfort due to direct interference with the vestibular system, which is responsible for balance. If discomfort is experienced, the VR evaluation may be paused until medical interventions are performed, if necessary. However, if discomfort persists, the participant will be removed from the research, the data collected will be excluded from analysis, and the patient will not be reallocated in another group.

To improve patient adherence to the intervention protocols, the patients will be contacted by phone to confirm they attended the treatment sessions, and reports of the results of the assessments that will be available.

The only interventions not permitted are those related to conventional VR and virtual VR.

This work was supported by the authors only.

## Quality assurance

The treatment protocol will be performed by two skilled researchers in the field of physical therapy/physical education who have previously performed this protocol inpatients with neurodegenerative diseases (spinocerebellar ataxia and Parkinson's disease) and will not be made aware of the results of the initial evaluations of the patients undergoingthe interventions.

## Assessment of results

The participants will be evaluated by an independent examiner who will be blinded to the allocations to the different groups. The following primary outcomes will be evaluated with each of the assessment tools, and improvements in performance from the pre intervention to the post intervention assessment will be assessed:

PPA score to assess the risk of falls;

Dynamometer findings to assess improvements in muscle strength;

WHOQOL-Bref score to assess improvements in perceived quality of life in different domains;

BBS score to assess balance status;

ABC to assess balance in terms of activities of daily living;

VADL score to investigate the impact of dizziness and balance and the degree of dependence;

MMSE score to measure the degree of cognitive impairment;

TUG test time to measure the risk of falls related to muscle weakness;

FES-I score to assess the fear of falling;

VAS score to quantify pain intensity.

The secondary outcomes will quantify the improvement in performance on the assessment. For such, a specific questionnaire will be administered for each assessment.

The first set of secondary outcomes will include locomotion capacity, stability during gait, and the magnitude of gait deviations, as evaluated using the TUG test and BBS.

The second set of secondary out comes will include spatial orientation capacity, as evaluated by the PPA, WHOQOL-Bref, ABC scale and MMSE.

The third set of secondary out comes will include the risk of falls, as evaluated by the FES-I, ABC scale, VADL scale and dynamometer.

The fourth set of secondary outcomes will include the capacity to perform activities of daily living, whether the level of self-confidence is sufficient to lead to positive gains in aspects related to family, social and professional life, as evaluated by the WHOQOL-Bref, VAS and VADL scale.

## Possible effects of treatment

The main clinical manifestation of HSP is a pyramidal pattern of slowly progressive weakness associated with lower limb weakness, which progresses to other manifestations, such as dementia, peripheral neuropathy, Parkinson's disease and ataxia.

The aim of VR with virtual reality is to delay the progression of the disease, providing a personalized, accessible, low-cost treatment option. The use of virtual reality with Nintendo Wii® is a novel, enjoyable, multisensory tool for balance training that promotes enthusiasm and motivation of the patient during continuous sessions. This technological resource can be used in addition to other conventional methods of VR.

## Data analysis

For data analysis, the difference between the pre- and post-therapy assessment times ($\Delta$ = T2—T1) will be calculated, thus enabling the comparison of the $\Delta$ values between the two groups (Group I and II) for each dependent variable. In addition, the normality of the distribution will be tested by the Shapiro-Wilk test. In case data normality occurs, Student's t test will be used to compare $\Delta$ values between groups, and a paired t-Student test for comparison within the group itself (pre- and post-). If the assumption of normality is not being met, non-parametric tests will be used: the Mann-Whitney test for independent samples (between groups) and the Wilcoxon test (Rank Sign Test) for dependent samples (within the group itself–pre- and post-). A 95% confidence interval will be used, and statistical significance will be reported for all differences between groups at $p$ <0.05. The results of the comparisons will be tabulated and box-plot graphs will be developed. All statistical analyzes will be performed on IBM SPSS for Windows (version 24.0, Armonk, NY. IBM. Corp.). A statistician blinded to group allocation will supervise the analysis. The sample size was calculated using the GPower® software (version 3.1.9.2) based on a previous study with a sample and similar therapy [12]. For the calculation, we used the average effect size observed in the performance obtained in vestibular rehabilitation with virtual reality by assuming a possible dropout rate of 10–20%, a total of 20 patients per group to reach error probability ($\alpha$) of 5%, with a power (1 –$\beta$ error probability) = 80%.

## Control of bias

The study includes several important methodological resources that will minimize the risk of bias, such as the randomization process, allocation concealment, blinded assessment of the results and intention-to-treat analysis.

## Timeline

Recruitment of the participants began in April 2019. All participants have been recruited, and treatment is expected to be completed by July 2021. The data analysis will be conducted in 2021, and the manuscript will be completed by December 2022.

## Conclusion

The study will provide a definitive assessment of the costs and benefits of VR programs involving virtual reality and their effectiveness in hindering the progression of the disease and serving as a personalized, accessible, low-cost treatment option. The use of virtual reality with Nintendo Wii® is an innovative, enjoyable, multisensory tool for balance training that promotes enthusiasm and motivation of the patient during continuous sessions. This technological resource can be used in addition to other conventional methods of VR.

## Supporting information

**S1 Checklist. CONSORT 2010 checklist of information to include when reporting a randomised trial.**
(DOC)

**S2 Checklist. SPIRIT 2013 checklist.**
(DOC)

**S1 File. Rebec.**
(DOCX)

## Acknowledgments

Thanks to the service of neurology at the hospital of the Federal University of Parana for giving permission for the accomplishment of the research and the Coordination for the Improvement of Higher Education Personnel—Brazil (CAPES).

## Author Contributions

**Conceptualization:** Bianca Simone Zeigelboim, Maria Renata José, Geslaine Janaina Bueno dos Santos, Hélio Afonso Ghizoni Teive, José Stechman-Neto, Rosane Sampaio Santos, Bianca Lopes Cavalcante-Leão.

**Formal analysis:** Cristiano Miranda de Araújo.

**Investigation:** Bianca Simone Zeigelboim, Maria Renata José, Geslaine Janaina Bueno dos Santos, Maria Izabel Rodrigues Severiano, Bianca Lopes Cavalcante-Leão.

**Methodology:** Bianca Simone Zeigelboim, Maria Renata José, Geslaine Janaina Bueno dos Santos, Maria Izabel Rodrigues Severiano, Hélio Afonso Ghizoni Teive, José Stechman-Neto, Rosane Sampaio Santos, Cristiano Miranda de Araújo, Bianca Lopes Cavalcante-Leão.

**Project administration:** Bianca Simone Zeigelboim, Maria Renata José, Geslaine Janaina Bueno dos Santos, Maria Izabel Rodrigues Severiano, Bianca Lopes Cavalcante-Leão.

**Resources:** Geslaine Janaina Bueno dos Santos, Bianca Lopes Cavalcante-Leão.

**Supervision:** Maria Renata José, José Stechman-Neto, Rosane Sampaio Santos, Bianca Lopes Cavalcante-Leão.

**Writing – original draft:** Bianca Simone Zeigelboim, Maria Renata José, Geslaine Janaina Bueno dos Santos, Maria Izabel Rodrigues Severiano, Cristiano Miranda de Araújo, Bianca Lopes Cavalcante-Leão.

**Writing – review & editing:** Bianca Simone Zeigelboim, Maria Renata José, Geslaine Janaina Bueno dos Santos, Maria Izabel Rodrigues Severiano, Hélio Afonso Ghizoni Teive, José Stechman-Neto, Rosane Sampaio Santos, Cristiano Miranda de Araújo, Bianca Lopes Cavalcante-Leão.

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
