## [Decision Letter · Decision Letter 0]

18 Dec 2020

PONE-D-20-32609

Balance rehabilitation with a virtual reality protocol for patients with hereditary spastic  paraplegia: Protocol for a clinical trial

PLOS ONE

Dear Dr. Cavalcante-Leão,

Thank you for submitting your manuscript to PLOS ONE. After careful consideration, we feel that it has merit but does not fully meet PLOS ONE’s publication criteria as it currently stands. Therefore, we invite you to submit a revised version of the manuscript that addresses the points raised during the review process.

We look forward to receiving your revised manuscript.

Kind regards,

Walid Kamal Abdelbasset, Ph.D.

Academic Editor

PLOS ONE

3. Please ensure that you include a title page within your main document. You should list all authors and all affiliations as per our author instructions and clearly indicate the corresponding author.

4. We note that this manuscript is a systematic review or meta-analysis; our author guidelines therefore require that you use PRISMA guidance to help improve reporting quality of this type of study. Please upload copies of the completed PRISMA checklist as Supporting Information with a file name “PRISMA checklist”.

Reviewer 1:

Thank you for giving the opportunity to review this article

Please edit the entire manuscript for English grammar and syntax for good presentation and readability.

Abstract:

1. Start with the subtitle - background

2. Mention clearly the duration of outcome measurement.

3. Mention the reports with 95% CI with p values.

4. Avoid abbreviations in the conclusion.

Introduction

1. The introduction part is too short and didn’t mention about important key points.

2. How come your study is differed from reference 9 and 10?

3. The research question is not formulated with suitable references.

4. Add more recent researches related virtual reality and its effects.

5. Define the clinical significance of this review in related to researchers, clinicians and patients.

Methods

6. Mention clearly who is diagnosing and selecting the patients for the study.

7. Make the inclusion and exclusion criteria in a paragraph format.

8. Missing of references for intervention procedures.

9. The reference for otolaryngolic evaluation and vestibular assessment.

10. The selection criteria should be more specific – (inclusion and exclusion)

11. Mention the method and referral study used for calculating the sample size.

Reviewer 2

The proposed randomized controlled clinical trial aims to determine the benefits of vestibular rehabilitation. Forty participants will be randomized to perform balance games (group I) or balance plus muscle strength games (group II). Patients will be assessed for changes in HSP after 10 weeks of intervention.

Minor revisions:

1. ABSTRACT: The abstract does not clearly indicate that the proposed study will be a randomized, controlled clinical trial. The methods section is composed with awkward language. For instance, the methods section contains verbs that are past tense while others are present tense.

2. Provide a more comprehensive statistical analysis plan. State the statistical methods that will be used to summarize the outcomes. Indicate the types of plots that will illustrate the results. Indicate that group I will be compared to group II using an ANOVA. Consider including an alternative analysis to ANOVA since the somewhat restrictive requirements and assumption of ANOVA may not be met. As an alternative consider a mixed linear model.

Reviewers' comments:

Reviewer's Responses to Questions

**Comments to the Author**

1. Does the manuscript provide a valid rationale for the proposed study, with clearly identified and justified research questions?

Reviewer #1: Yes

Reviewer #2: Yes

2. Is the protocol technically sound and planned in a manner that will lead to a meaningful outcome and allow testing the stated hypotheses?

Reviewer #1: Yes

Reviewer #2: Yes

3. Is the methodology feasible and described in sufficient detail to allow the work to be replicable?

Reviewer #1: Yes

Reviewer #2: Yes

4. Have the authors described where all data underlying the findings will be made available when the study is complete?

Reviewer #1: No

Reviewer #2: Yes

5. Is the manuscript presented in an intelligible fashion and written in standard English?

Reviewer #1: Yes

Reviewer #2: Yes

6. Review Comments to the Author

You may also provide optional suggestions and comments to authors that they might find helpful in planning their study.

Reviewer #1: Reviewer comments:

Thank you for giving the opportunity to review this article

Please edit the entire manuscript for English grammar and syntax for good presentation and readability.

Abstract:

1. Start with the subtitle - background

2. Mention clearly the duration of outcome measurement.

3. Mention the reports with 95% CI with p values.

4. Avoid abbreviations in the conclusion.

Introduction

1. The introduction part is too short and didn’t mention about important key points.

2. How come your study is differed from reference 9 and 10?

3. The research question is not formulated with suitable references.

4. Add more recent researches related virtual reality and its effects.

5. Define the clinical significance of this review in related to researchers, clinicians and patients.

Methods

6. Mention clearly who is diagnosing and selecting the patients for the study.

7. Make the inclusion and exclusion criteria in a paragraph format.

8. Missing of references for intervention procedures.

9. The reference for otolaryngolic evaluation and vestibular assessment.

10. The selection criteria should be more specific – (inclusion and exclusion)

11. Mention the method and referral study used for calculating the sample size.

Reviewer #2: The proposed randomized controlled clinical trial aims to determine the benefits of vestibular rehabilitation. Forty participants will be randomized to perform balance games (group I) or balance plus muscle strength games (group II). Patients will be assessed for changes in HSP after 10 weeks of intervention.

Minor revisions:

1- ABSTRACT: The abstract does not clearly indicate that the proposed study will be a randomized, controlled clinical trial. The methods section is composed with awkward language. For instance, the methods section contains verbs that are past tense while others are present tense.

2- Provide a more comprehensive statistical analysis plan. State the statistical methods that will be used to summarize the outcomes. Indicate the types of plots that will illustrate the results. Indicate that group I will be compared to group II using an ANOVA. Consider including an alternative analysis to ANOVA since the somewhat restrictive requirements and assumption of ANOVA may not be met. As an alternative consider a mixed linear model.

7. PLOS authors have the option to publish the peer review history of their article (what does this mean?). If published, this will include your full peer review and any attached files.

Reviewer #2: No

---

## [Author Response · Author response to Decision Letter 0]

26 Jan 2021

Dear Sir,

Thanks for the reviewer's comments about our article. All corrections were made and the questions answered.

REVISION NOTES

REVIEWER 1

Abstract:

1. Start with the subtitle – background All suggestions have been considered and changed in the text.

2. Mention clearly the duration of outcome measurement. All suggestions have been considered and changed in the text. (line 385-387).

3. Mention the reports with 95% CI with p values. All suggestions have been considered and changed in the text.

4. Avoid abbreviations in the conclusion. All suggestions have been considered and changed in the text. 

Introduction

1. The introduction part is too short and didn’t mention about important key points. All suggestions have been considered and changed in the text.

2. How come your study is differed from reference 9 and 10? The present study deals with hereditary spastic paraplegia, a degenerative disease involving the central nervous system and this differs from the references cited [9 and 10] in that different diseases are evaluated and different equipment is used in virtual reality rehabilitation. In the references cited, chronic diseases and Ménière's disease, determined by tinnitus, vertigo and hearing loss, were evaluated in paroxysmal episodes without involvement of the central nervous system, with endolymphatic hydrops being considered as the pathophysiological basis of this disease. The equipment used was the Balance Rehabilitation Unit (BRUTM).

3. The research question is not formulated with suitable references.

The objective of this study as well as the question is to determine the benefits of VR involving virtual reality, comparing the results of pre- and post-intervention evaluations in individuals with hereditary spastic paraplegia, as the great motivation for carrying out this study was based on the fact there are no studies like the one mentioned in the introduction. "No randomized or nonrandomized studies related to VR with virtual reality in patients with HSP have been published in the PubMed electronic databases, including Medline, Scielo, Embase, Scopus and Web of Science".

4. Add more recent researches related virtual reality and its effects. All suggestions have been considered and changed in the text.

5. Define the clinical significance of this review in related to researchers, clinicians and patients. This research is a clinical trial not a review. 

Methods

6. Mention clearly who is diagnosing and selecting the patients for the study. All suggestions have been considered and changed in the text.

7. Make the inclusion and exclusion criteria in a paragraph format. All suggestions have been considered and changed in the text.

8. Missing of references for intervention procedures. References 2, 4, 10, 11, 12, 13 and 14 include intervention procedures with virtual reality.

9. The reference for otolaryngolic evaluation and vestibular assessment. All suggestions have been considered and changed in the text.

10. The selection criteria should be more specific – (inclusion and exclusion) All suggestions have been considered and changed in the text.

11. Mention the method and referral study used for calculating the sample size. All suggestions have been considered and changed in the text.

(line 373).

REVIEWER 2

Minor revisions:

1. ABSTRACT: The abstract does not clearly indicate that the proposed study will be a randomized, controlled clinical trial. The methods section is composed with awkward language. For instance, the methods section contains verbs that are past tense while others are present tense. All suggestions have been considered and changed in the text.

2. Provide a more comprehensive statistical analysis plan. State the statistical methods that will be used to summarize the outcomes. Indicate the types of plots that will illustrate the results. Indicate that group I will be compared to group II using an ANOVA. Consider including an alternative analysis to ANOVA since the somewhat restrictive requirements and assumption of ANOVA may not be met. As an alternative consider a mixed linear model.

All suggestions have been considered and changed in the text.

---

## [Decision Letter · Decision Letter 1]

1 Mar 2021

PONE-D-20-32609R1

Balance rehabilitation with a virtual reality protocol for patients with hereditary spastic  paraplegia: Protocol for a clinical trial

PLOS ONE

Dear Dr. Cavalcante-Leão,

Thank you for submitting your manuscript to PLOS ONE. After careful consideration, we feel that it has merit but does not fully meet PLOS ONE’s publication criteria as it currently stands. Therefore, we invite you to submit a revised version of the manuscript that addresses the points raised during the review process.

We look forward to receiving your revised manuscript.

Kind regards,

Walid Kamal Abdelbasset, Ph.D.

Academic Editor

PLOS ONE

Journal Requirements:

Reviewers' comments:

Reviewer's Responses to Questions

**Comments to the Author**

1. Does the manuscript provide a valid rationale for the proposed study, with clearly identified and justified research questions?

Reviewer #1: Yes

Reviewer #2: Yes

2. Is the protocol technically sound and planned in a manner that will lead to a meaningful outcome and allow testing the stated hypotheses?

Reviewer #1: Yes

Reviewer #2: Yes

3. Is the methodology feasible and described in sufficient detail to allow the work to be replicable?

Reviewer #1: Yes

Reviewer #2: Yes

4. Have the authors described where all data underlying the findings will be made available when the study is complete?

Reviewer #1: Yes

Reviewer #2: Yes

5. Is the manuscript presented in an intelligible fashion and written in standard English?

Reviewer #1: Yes

Reviewer #2: Yes

6. Review Comments to the Author

You may also provide optional suggestions and comments to authors that they might find helpful in planning their study.

Reviewer #1: Reviewer comments:

Thank you for giving the opportunity to review this article.

1. Results part should be more informative including CI 95% with p values.

2. Present the article in a simple past tense than future tense.

3. Include latest references in the field of virtual reality training.

4. Please provide the definitive conclusion.

Reviewer #2: I have no additional comments.

7. PLOS authors have the option to publish the peer review history of their article (what does this mean?). If published, this will include your full peer review and any attached files.

Reviewer #1: **Yes: **Gopal Nambi

Reviewer #2: No

---

## [Author Response · Author response to Decision Letter 1]

3 Mar 2021

March, 1st.

Dear Sir,

Thanks for the reviewer's comments about our article. All corrections were made and the questions answered.

REVISION NOTES

REVIEWER 1

1. Results part should be more informative including CI 95% with p values. Thank you for your comment, but we need to inform you that the results will be informed after the execution of the protocol of this clinical trial. 

2. Present the article in a simple past tense than future tense. The text in general is in the future because it is a clinical trial protocol.

3. Include latest references in the field of virtual reality training. All suggestions have been considered and made the reference insertion number 10.

4. Please provide the definitive conclusion. In the same way that we do not have the results definitively, we also do not provide a definitive conclusion. Because this is a trial protocol.

REVIEWER 2

I have no additional comments. Thank you for your comments.

---

## [Decision Letter · Decision Letter 2]

11 Mar 2021

Balance rehabilitation with a virtual reality protocol for patients with hereditary spastic  paraplegia: Protocol for a clinical trial

PONE-D-20-32609R2

Dear Dr. Cavalcante-Leão,

We’re pleased to inform you that your manuscript has been judged scientifically suitable for publication and will be formally accepted for publication once it meets all outstanding technical requirements.

Kind regards,

Walid Kamal Abdelbasset, Ph.D.

Academic Editor

PLOS ONE

Additional Editor Comments (optional):

All required corrections are completely addressed by the authors

Reviewers' comments:

Reviewer's Responses to Questions

**Comments to the Author**

1. Does the manuscript provide a valid rationale for the proposed study, with clearly identified and justified research questions?

Reviewer #1: Yes

Reviewer #2: Yes

2. Is the protocol technically sound and planned in a manner that will lead to a meaningful outcome and allow testing the stated hypotheses?

Reviewer #1: Yes

Reviewer #2: Yes

3. Is the methodology feasible and described in sufficient detail to allow the work to be replicable?

Reviewer #1: Yes

Reviewer #2: Yes

4. Have the authors described where all data underlying the findings will be made available when the study is complete?

Reviewer #1: Yes

Reviewer #2: Yes

5. Is the manuscript presented in an intelligible fashion and written in standard English?

Reviewer #1: Yes

Reviewer #2: Yes

6. Review Comments to the Author

You may also provide optional suggestions and comments to authors that they might find helpful in planning their study.

Reviewer #1: Authors have satisfactorily justified the comments raised by me. Hence the article can be accepted in the present format.

Reviewer #2: I have no additional comments.

7. PLOS authors have the option to publish the peer review history of their article (what does this mean?). If published, this will include your full peer review and any attached files.

Reviewer #1: **Yes: **Gopal Nambi

Reviewer #2: No

---

## [Editor Report · Acceptance letter]

15 Mar 2021

PONE-D-20-32609R2 

Balance rehabilitation with a virtual reality protocol for patients with hereditary spastic paraplegia: Protocol for a clinical trial 

Dear Dr. Cavalcante-Leão:

I'm pleased to inform you that your manuscript has been deemed suitable for publication in PLOS ONE. Congratulations! Your manuscript is now with our production department. 

Kind regards, 

on behalf of

Dr. Walid Kamal Abdelbasset 

Academic Editor

PLOS ONE